 **eLIFE**

# Logics and properties of a genetic regulatory program that drives embryonic muscle development in an echinoderm

Carmen Andrikou[1†], Chih-Yu Pai[2], Yi-Hsien Su[2]*, Maria Ina Arnone[1]*

[1]Biology and Evolution of Marine Organisms, Stazione Zoologica Anton Dohrn, Napoli, Italy; [2]Institute of Cellular and Organismic Biology, Academia Sinica, Taipei, Taiwan

**Abstract** Evolutionary origin of muscle is a central question when discussing mesoderm evolution. Developmental mechanisms underlying somatic muscle development have mostly been studied in vertebrates and fly where multiple signals and hierarchic genetic regulatory cascades selectively specify myoblasts from a pool of naive mesodermal progenitors. However, due to the increased organismic complexity and distant phylogenetic position of the two systems, a general mechanistic understanding of myogenesis is still lacking. In this study, we propose a gene regulatory network (GRN) model that promotes myogenesis in the sea urchin embryo, an early branching deuterostome. A fibroblast growth factor signaling and four Forkhead transcription factors consist the central part of our model and appear to orchestrate the myogenic process. The topological properties of the network reveal dense gene interwiring and a multilevel transcriptional regulation of conserved and novel myogenic genes. Finally, the comparison of the myogenic network architecture among different animal groups highlights the evolutionary plasticity of developmental GRNs.

*For correspondence: yhsu@ gate.sinica.edu.tw (Y-HS); miarnone@szn.it (MIA)

Present address: †Sars International Centre for Marine Molecular Biology, University of Bergen, Bergen, Norway

Competing interests: The authors declare that no competing interests exist.

## Introduction

Muscle cells are present in most animals with the characteristic of containing protein filaments that slide and produce contractions. In bilaterians, muscles develop usually from mesodermal cells in a process called myogenesis, during which the naive mesodermal progenitors are selectively specified as myoblasts and later differentiate into muscle cells. In vertebrate embryos, the mesoderm is subdivided into several regions from which different muscle types originated. For example, the axial skeletal muscles originate in the segmented paraxial mesoderm called somites, whereas the cardiac muscles develop from the lateral plate mesoderm. The subdivisions of different mesoderm regions require differential expression of the Forkhead (Fox) family of transcription factors, such as FoxC1 and FoxC2 for the paraxial mesoderm (*Wilm et al., 2004*) and FoxF1 for the lateral plate mesoderm (*Mahlapuu et al., 2001*). Analyses of molecular mechanisms underlying myogenesis in various parts of the somites further lead to a series of complex gene regulatory networks (GRNs) (*Bryson-Richardson and Currie, 2008*; *Yokoyama and Asahara, 2011*; *Bentzinger et al., 2012*). These networks are composed of multiple signals that drive the expression of the genes encoding basic helix-loop-helix (bHLH) domain-containing myogenic regulatory factors, which include myogenic differentiation 1 (MyoD), myogenic factor 5 (Myf5), Myf6, and myogenin (Myog). Other factors, including the myocyte enhancer binding factor 2 (MEF2) family of MADS-box proteins, transcription factors Pitx2, Pitx3, sine oculis-related homeobox (six) family members and their cofactors eyes-absent homologs (Eya), are also involved in specification and migration of the myogenic precursor cells (*Molkentin and Olson, 1996*; *Yokoyama and Asahara, 2011*). These myogenic transcriptional regulators then in turn initiate the transcription of muscle differentiation markers, such as myosin heavy chain proteins (MHC), that determine the morphological

**eLife digest** Muscles, bones, and blood vessels all develop from a tissue called the mesoderm, which forms early on in the development of an embryo. Networks of genes control which parts of the mesoderm transform into different cell types. The gene networks that control the development of muscle cells from the mesoderm have so far been investigated in flies and several species of animals with backbones. However, these species are complex, which makes it difficult to work out the general principles that control muscle cell development.

Sea urchins are often studied in developmental biology as they have many of the same genes as more complex animals, but are much simpler and easier to study in the laboratory. Andrikou et al. therefore investigated the 'gene regulatory network' that controls muscle development in sea urchins. This revealed that proteins called Forkhead transcription factors and a process called FGF signaling are crucial for controlling muscle development in sea urchins. These are also important factors for developing muscles in other animals.

Andrikou et al. then produced models that show the interactions between the genes that control muscle formation at three different stages of embryonic development. These models reveal several important features of the muscle development gene regulatory network. For example, the network is robust: if one gene fails, the network is connected in a way that allows it to still make muscle. This also allows the network to adapt and evolve without losing the ability to perform any of its existing roles.

Comparing the gene regulatory network that controls muscle development in sea urchins with the networks found in other animals showed that many of the same genes are used across different species, but are connected into different network structures. Investigating the similarities and differences of the regulatory networks in different species could help us to understand how muscles have evolved and could ultimately lead to a better understanding of the causes of developmental diseases.

and functional identity of the differentiated muscle cells (*Bryson-Richardson and Currie, 2008*; *Bentzinger et al., 2012*).

In addition to the transcription factors mentioned above, myogenesis in different parts of the vertebrate somites depends on several signals including Notch, fibroblast growth factor (FGF), Wnts, bone morphogenetic factor 4 (BMP4), and Sonic hedgehog (Shh) that are secreted from adjacent tissues, such as neural tube, notochord, dorsal, and lateral ectoderms (*Marcelle et al., 1997*; *Vasyutina et al., 2007*; *Delfini et al., 2009*). Among these signals, FGF-signaling pathway is of particular interest because it has a crucial and conserved role in mesoderm patterning and muscle development in various animal models. In vertebrates, FGF induces the expression of myogenic genes in the primary myotome, controls the timing of the epithelial–mesenchymal transition of the dermomyotome, which triggers the emergence of muscle progenitors (*Delfini et al., 2009*) and positively regulates muscle differentiation (*Marics et al., 2002*; *Groves et al., 2005*). In flies, an FGFR ortholog (*Heartless*)-mediated pathway is essential for cell migration and fate induction of the visceral mesoderm, heart, and the somatic muscle lineages (*Beiman et al., 1996*). In ascidians, FGF signaling is required for the specification of the heart progenitor cells and secondary muscle development (*Beh et al., 2007*; *Tokuoka et al., 2007*). In the nematode *Caenorhabditis elegans*, the crosstalk between FGF and Wnt signals is involved in larval sex myoblast specification and migration (*Burdine et al., 1998*; *Lo et al., 2008*). In the amphioxus *Brachiostoma lanceolatum*, FGF signaling is necessary for the formation of the anterior somites that will generate the musculature (*Bertrand et al., 2011*) and in the hemichordate *Saccoglossus kowalevskii*, FGF signaling is necessary for all types of mesoderm development (*Green et al., 2013*).

The overall genetic regulatory cascade that drives myogenesis appears to be highly conserved between ecdysozoans (*Drosophila melanogaster* and *C. elegans*) and vertebrates (*Ciglar and Furlong, 2009*). However, due to the high complexity that characterizes the vertebrate organogenesis and the distant phylogenetic position between ecdysozoans and vertebrates, the logics behind the genomic regulatory interactions that drive muscle development remain unclear. Therefore, elucidating the properties of the key genetic interconnections that orchestrate myogenesis in a larger set of model

organisms is necessary to reveal the core myogenic circuits. Moreover, given the fact that the common origin of musculature is a highly debated topic (*Seipel and Schmid, 2005*; *Burton, 2008*; *Steinmetz et al., 2012*), a larger interspecies comparison of the transcriptional myogenic networks would provide new insights into the evolution of muscle and the kernel driven hypothesis.

Echinoderms occupy a key phylogenetic position since they are early branching deuterostomes, therefore, are more closely related to vertebrates compared to the other two most studied invertebrate model systems (*D. melanogaster* and *C. elegans*). Sea urchin is a powerful model system for studying regulatory events that take place during development due to their advantageous properties for GRN analysis (*Oliveri and Davidson, 2004*). GRNs provide causal explanations of the molecular interactions occurring during dynamic developmental processes such as cell specification and differentiation (*Davidson et al., 2002*; *Ben-Tabou de-Leon and Davidson, 2006*). In the last decade, the extensive amount of data collected by perturbation analyses has led to the assembly of the largest so far known endomesodermal GRN (*Davidson et al., 2002*; *Peter and Davidson, 2011*; *Materna et al., 2013*) representing most of the transcriptional interactions that take place during endomesoderm formation in the sea urchin embryo. These properties make echinoderms an excellent model to study molecular developmental mechanisms and address aspects in evolution of organogenesis.

Sea urchin larvae possess a muscular apparatus that surrounds their esophagus and produces contractile force (*Burke, 1981*; *Burke and Alvarez, 1988*). These muscle cells originate from mesoderm and adapt the myogenic fate by segregating from the other three non-skeletogenic mesodermal (NSM) lineages at the early gastrula stage (*Ruffins and Ettensohn, 1996*). The first appearance of myoblasts is seen at the late gastrula stage as a few cells in the oral vegetal domain of each coelomic sac at the tip of the archenteron and they express a number of muscle-specific transcription regulators (*Andrikou et al., 2013*). Few hours later, at the prism stage, these cells extend pseudopods towards the midline of the esophagus, increase in number and diameter, fuse to each other, and finally form the circumesophageal contractile bands (*Burke and Alvarez, 1988*).

Despite the existence of large amount of data concerning the mechanisms patterning the early endomesoderm segregation, little is known about the regulatory landscape of the later NSM specification. Parts of the regulatory events that underlie the specification of the three NSM mesodermal lineages (blastocoelar, pigment, and coelomic pouch cell lineages) were recently revealed (*Luo and Su, 2012*; *Ransick and Davidson, 2012*; *Materna et al., 2013*; *Solek et al., 2013*), and only scattered information regarding the molecular interplay that establish the muscle lineage is available in the literature. In particular, transcription factors such as Twist (*Wu et al., 2008*) and a sea urchin lineage-specific Fox family factor, FoxY (*Materna et al., 2013*), and signaling pathways such as Delta/Notch (D/N) (*Sherwood and McClay, 1999*; *Sweet et al., 2002*) and Hedgehog (Hh) (*Walton et al., 2009*; *Warner et al., 2014*) seem to be involved in muscle development, but the molecular mechanisms remain poorly understood.

In our previous work, we identified several homologues of myogenic regulators in the sea urchin embryo and revealed the regulatory state of the myoblasts and their precursor cells (*Andrikou et al., 2013*). Further functional analyses are needed to confirm the myogenic function of these transcription factors and to unravel the regulatory architecture of the muscle GRN. This study provides an explanatory mechanism on how the sea urchin muscle lineage is specified using a perturbation approach accompanied by a combination of temporal and spatial gene expression analyses. We show that FGF signaling is necessary for specifying naive NSM cells to myoblast precursors at the very early gastrula stage. The four Fox family factors FoxY, FoxC, FoxF, and FoxL1 (*Tu et al., 2006*) constitute the central part of our myogenic GRN model and are activated sequentially. Other conserved key factors that occupy different hierarchical levels in the myogenic GRN are members of the T-box (Tbx6) (*Howard-Ashby et al., 2006*), bHLH (MyoD2) (*Andrikou et al., 2013*), SRY (Sox) (*Howard-Ashby et al., 2006*), Scratch (ScratchX) (*Materna et al., 2013*), and Six (Six1/2) (*Poustka et al., 2007*) family genes. These findings imply an overall high level of functional conservation of both key myogenic transcriptional regulators and signaling components but not of the myogenic GRN architecture per se. Also, they explain in a rational way the logics and properties of the regulatory interactions that drive myogenesis in the sea urchin embryo.

## Results

### Genes encoding FGF-signaling components are expressed in the putative myoblast precursors

As shown in our previous work (*Andrikou et al., 2013*), the myogenic lineage seems to segregate as early as the very early gastrula stage (30 hr post fertilization). The relative position of these cells is not precise at the earlier stages but they seem to be located at the oral/lateral periphery of the vegetal plate, at the border between the blastocoelar and pigment cell precursors (*Figure 1—figure supplement 1* [*Ruffins and Ettensohn, 1996*]). To reveal the cause of myoblast emergence at the early gastrula stage, we searched for putative candidate signaling components that are expressed at that time in the myoblast precursors. As elsewhere demonstrated, a mesodermal Delta/Notch signal activates *FoxY*, which is required for the specification of the two mesoderm derivatives, coelomic pouches and muscles (*Materna and Davidson, 2012*; *Materna et al., 2013*). The segregation of these two derivatives should thus specifically rely on another signal that triggers myoblast specification at that particular developmental time.

There are one FGF ligand, FGFA (*Figure 1—figure supplement 2*), and two FGF Receptors (FGFRs), FGFR1, and FGFR2 (*Figure 1—figure supplement 3*), annotated in the sea urchin genome (*McCoon et al., 1996*; *Lapraz et al., 2006*). A previous study showed that inhibition of FGFR2 affects morphogenesis of the embryonic skeleton (*Rottinger et al., 2008*). We therefore focused on characterizing the expression patterns of genes encoding FGFR1 and FGFA ligand in the sea urchin *Strongylocentrotus purpuratus*. The expression profile of both genes matches with the one already described in another sea urchin species, *Paracentrotus lividus* (*Rottinger et al., 2008*). To better resolve the expression patterns of these two genes relatively to the myoblast precursors, double fluorescent in situ hybridizations (FISHs) with *FoxC*, the earliest myoblast precursor marker (*Andrikou et al., 2013*), were performed. At the very early gastrula stage, *FGFR1* was expressed in all *FoxC*-positive cells (*Figure 1A*) whilst at the late gastrula stage (48 hr), only a number of *FGFR1* transcripts overlap with *FoxC*-positive cells at the tip of the archenteron (*Figure 1—figure supplement 4*). Moreover, *FGFA* transcripts were observed in the ventrolateral ectodermal regions in the vicinity of *FoxC*-positive cells (*Figure 1B*). The spatial and temporal expression of *FGFR1* and *FGFA* suggests a putative role of FGF signaling in sea urchin myogenesis.

### FGF signaling specifies the time of the myoblast emergence and acts through a MAPK/ERK cascade

To test whether FGF signaling through FGFR1 is involved in sea urchin myogenesis, we perturbed FGF-signaling pathway by using (a) SU5402, a FGFR inhibitor (*Mohammadi et al., 1997*); (b) U0126, a MEK inhibitor (*Favata et al., 1998*); (c) an antisense morpholino oligonucleotide (MO) targeted to FGFR1; (d) a dominant negative form (Dn) of FGFR1. A summary of the phenotypes observed after inhibitor treatments is in *Figure 2—figure supplement 1A*. 70% of the 300 pluteus larvae treated with SU5402 from 26 hr showed an abnormal elongated archenteron missing the pyloric and possibly the anal sphincter constrictions, although the cardiac sphincter was still formed (*Figure 2A–D*) suggesting that the three distinct sphincters are differentially regulated (*Annunziata and Arnone, 2014*). Moreover, all larvae had shorter spicules and their triradiate skeleton was never fully shaped possibly due to the inhibition of FGFR2, which is specifically expressed in the skeletogenic primary mesenchyme cells (PMCs) that produce the larval skeleton (*Rottinger et al., 2008*). Finally, the coelomic pouches were not formed and the circumesophagael muscle fibers were completely absent when tested for *MHC* gene expression by FISH at the prism stage (*Figure 2D*). To reveal the downstream cascade of the FGF signaling, we used the MEK inhibitor U0126. In one of our previous studies, we showed that MAPK/ERK activation is required for muscle formation (*Fernandez-Serra et al., 2004*). We repeated the experiment by treating the embryos with U0126 at 26 hr, as we did with the SU5402 inhibitor. The treated larvae were tested again for the presence of muscle fibers by examining the MHC protein level using immunostaining. Again, the level of MHC protein was greatly reduced in the treated larvae (*Figure 2—figure supplement 1B,C*). These findings show an involvement of FGF signaling through MAPK/ERK pathway in muscle development.

To reinforce the role of FGF signaling mediated through FGFR1 during myogenesis, we performed gene knockdown experiments by injecting anti-translation MO against FGFR1 at

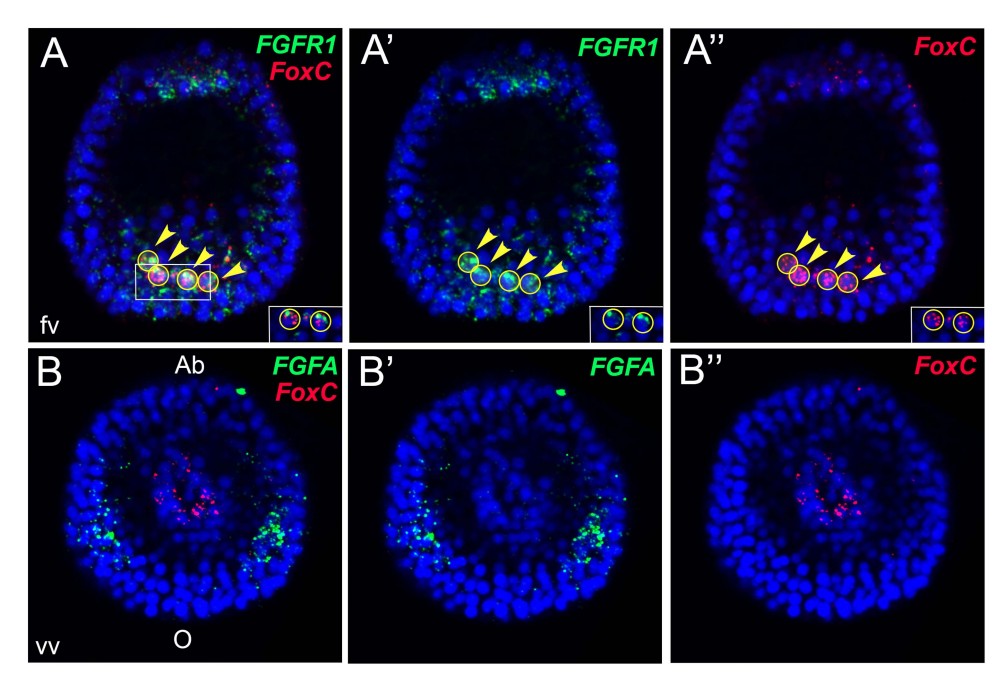

**Figure 1**. Expression analysis of genes encoding sea urchin FGF-signaling components and *FoxC* by double FISH. *FGFR1* and *FGF* were stained in green and *FoxC* in red at the very early gastrula stage (30–32 hr). Nuclei were labeled blue with DAPI. Yellow circles indicated by yellow arrowheads show cells co-expressing the analyzed genes. Panels **A** and **B** are stacks of merged confocal Z sections of all three channels, while separate channels over DAPI are presented in the other panels. Insets in panels **A**–**A″** show representative single confocal sections to confirm that the two genes are indeed expressed in the same cell. Embryos in **A**–**A″** are seen in a lateral view along the animal-top/vegetal-down axis. Embryos in **B**–**B″** are displayed in a vegetal view. fv, frontal view; vv, vegetal view; o, oral, ab, aboral. The position of the putative unspecified myoblast precursors is indicated in *Figure 1—figure supplement 1*. Phylogenetic analyses of sea urchin fibroblast growth factor (FGF) and FGFR protein sequences are reported in *Figure 1—figure supplements 2, 3*, respectively. A co-expression analysis of *FGFR1* and *FoxC* at late gastrula stage (48 hr) is reported in *Figure 1—figure supplement 4*.

The following figure supplements are available for figure 1:

**Figure supplement 1**. Three-color FISH of *Gcm, Ese*, and *FoxA*.

**Figure supplement 2**. Phylogenetic analysis of the sea urchin FGFA protein sequence.

**Figure supplement 3**. Phylogenetic analysis of the sea urchin FGFR protein sequences.

**Figure supplement 4**. Coexpression analysis of *FGFR1* and *FoxC* by double FISH.

---

a concentration of 500 μM. The MO toxicity and efficacy were examined by using a control MO and a GFP fusion construct, respectively (*Figure 2—figure supplement 2*). Although in 90% of the morphant larvae, a malformation of coelomic pouches (*Figure 2F*) and a severe down-regulation of the MHC protein level were evident (*Figure 2G,H*), two different phenotypes concerning the compartmentalization of the archenteron were observed. 70% of the larvae displayed a fully elongated gut, with the cardiac and anal sphincters well formed (*Figure 2F*), while in the rest cases both sphincters were absent or partly formed (*Figure 2H*). In addition, an abnormal extension of cilia in the apical ectoderm was observed (*Figure 2H* insert) that provides a putative additional role of FGFR1 in patterning the apical organ (Paola Oliveri, personal communication) consistent with the expression of *FGFR1* in the apical domain (*Figure 1A*). An evident reduction of the cilia in the gut lumen is also observed (*Figure 2H*), which may be related to the FGFR1 expression seen in

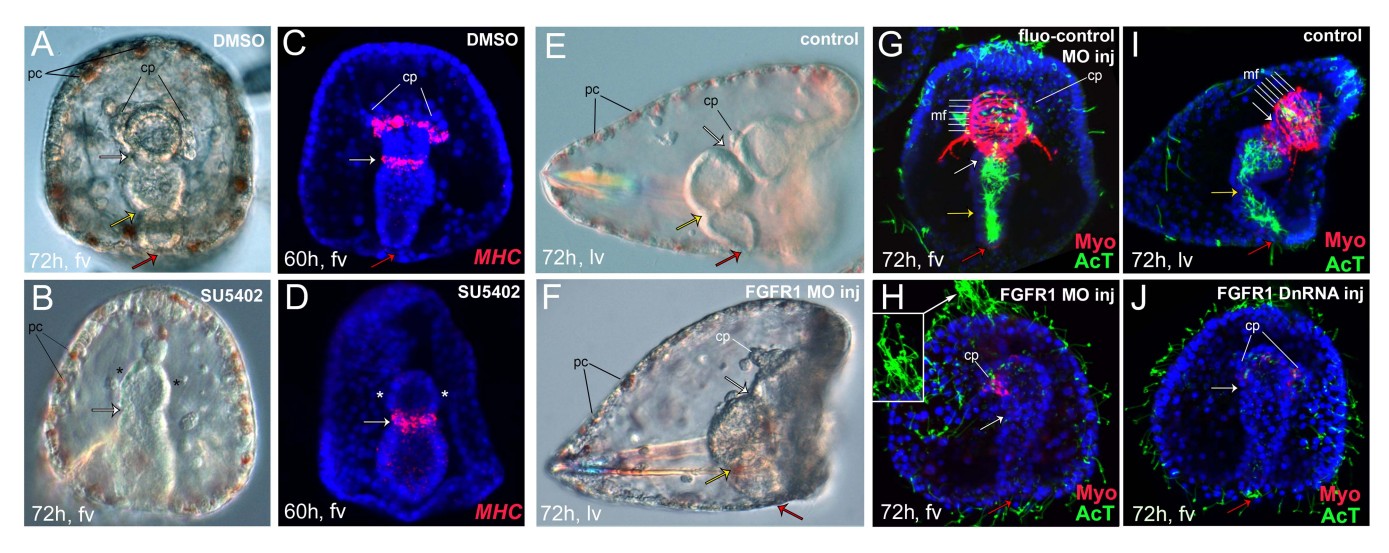

**Figure 2**. Perturbation of the FGF pathway. To analyze the phenotype of FGF perturbation, bright-field images were taken with differential interference contrast (DIC). Effects on muscle formation were also tested by detection of *MHC* expression by fluorescent in situ hybridization (FISH) or of myosin heavy chain (MHC) protein localization by immunostaining on pluteus larvae (72 hr). The ciliary band and gut internal cilia were stained with an anti-acetylated tubulin antibody (AcT). Panels (**A**–**D**) show the effect of SU5402 in the formation of the coelomic pouches (**B**) and *MHC* expression (**D**). Panels (**E**–**H**) show the effect of anti-FGFR1 translation morpholino oligonucleotide (MO) in the formation of the coelomic pouches, MHC protein localization, and gut morphology. Two representative phenotype embryos, both with impaired muscles while differing for gut sphincter formation, are reported in **F** (normal gut, 70% of cases) and **H** (reduced sphincters, 30% of cases). Panels (**E**, **G**, **I**, and **J**) show the effect in MHC protein localization caused by injection of FGFR1 dominant negative RNA (DnRNA) (**J**). Panels (**A**, **C**) show control embryos treated with DMSO. Panel (**G**) shows a larva injected with a fluo-control MO and panels (**E**, **I**) show control uninjected larvae (for MO injection controls see also 'Materials and methods' and *Figure 2—figure supplement 2*). The inset in panel H is a magnified view of the cilia at the apical organ. Pictures in **C**, **D**, and **G**–**J** are stacks of merged confocal Z sections. MHC was stained in red and acetylated tubulin in green. Nuclei were labeled blue with DAPI. Spicules are seen in DIC analysis as reflecting polarized light objects. All embryos are seen in frontal view except the ones in panels **E**, **F**, and **I** that are seen in lateral view with the oral side on the right (fv, frontal view; lv, lateral view). White arrows indicate the position of cardiac sphincters, whilst yellow and red arrows show, where present, the pyloric and anal sphincters, respectively. Black lines indicate pigment cells (pc). White lines indicate muscle fibers (mf). The asterisks indicate the absence of coelomic pouches (cp). A summary of SU5402 and U0126 treatments as well as MHC protein expression analysis after MEK pathway perturbation is reported in *Figure 2—figure supplement 1*. Control MO experiments are reported in *Figure 2—figure supplement 2*. Co-expression analysis of genes encoding putative MAPK effectors and *FoxC* as well as P-Elk protein detection is reported in *Figure 2—figure supplement 3*.

The following figure supplements are available for figure 2:

**Figure supplement 1**. Summary of SU5402 and U0126 treatments and MHC protein detection by immunostaining after MEK pathway perturbation.

**Figure supplement 2**. Control experiments for MOs.

**Figure supplement 3**. Immunostaining of P-Elk and expression analysis of genes encoding putative MAPK effectors and *FoxC* by double FISH.

the endoderm during gastrulation (*Figure 1—figure supplement 4*). Moreover, the larval skeleton and the formation of pigment cells were not affected, confirming that FGFR1 is not involved either in PMC patterning and subsequent skeleton formation or in pigment cell development (*Figure 2F*). An even more severe phenotype was obtained by injecting the embryos with the mRNA encoding FGFR1 Dn with the cardiac and anal sphincters being absent in 60% of the cases (*Figure 2J*). As expected, MHC levels were greatly reduced whilst the same abnormal extension of cilia in the apical ectoderm and the reduction of the cilia in the gut lumen were observed. Taking together, these experiments not only demonstrate clearly an essential role of FGF signaling through FGFR1 in sea urchin myogenesis but also suggest its possible involvement in the formation of the ciliated gut epithelium and the ciliary ectoderm.

We then tested the expression of the myoblast marker genes *FoxY*, *FoxC*, and *FoxF* in the FGF signaling-perturbed embryos at the very early gastrula (28 hr), mid gastrula (36 hr), and late gastrula

(48 hr) stages, respectively. While *FoxY* expression did not change when FGF signaling was perturbed (*Figure 3A–D*), a severe impact was observed in *FoxC* and *FoxF* transcript levels in all treated embryos (*Figure 3E–L*). To check whether the specification of other oral NSM cell lineages were affected as well by the FGFR1 perturbation, we also tested the expression of *Ese*, a marker of the blastocoelar cell lineage, and found it unaffected (*Figure 3A,B*). These results further support the involvement of FGF signaling in myogenesis and demonstrate that *FoxC* and *FoxF* factors are acting downstream of it.

ETS family transcription factors are known effectors of the MAPK-signaling pathway. In order to understand whether ETS-domain-containing transcriptional regulators are likely to be the downstream effector of the MAPK pathway leading to *FoxC* and *FoxF* transcription in presumptive myoblasts, we searched for ETS family factors that are significantly expressed in the NSM at the very early gastrula stage. From the 11 members of the ETS gene family that are present in the sea urchin genome, only 3 are expressed in the NSM at 30 hr; *Elk*, *Erg*, and *Ets1/2* (*Rizzo et al., 2006*). We performed an immunostaining experiment on the phosphorylated active form of Elk (P-Elk) and double FISH of *FoxC* and *Erg* or *Ets1/2*. The spatial localization of P-Elk protein seemed to be excluded from the myogenic cells, when comparing it to the expression pattern of *FoxC* (*Figure 2—figure supplement 3A–C*), suggesting that P-Elk is probably not the candidate effector of MAPK involved in myoblast specification. *Erg* expression was also not coincided with *FoxC* transcripts at 30 hr (*Figure 2—figure supplement 3C,D*), thus, indicating that this factor is also not the downstream effector. On the other

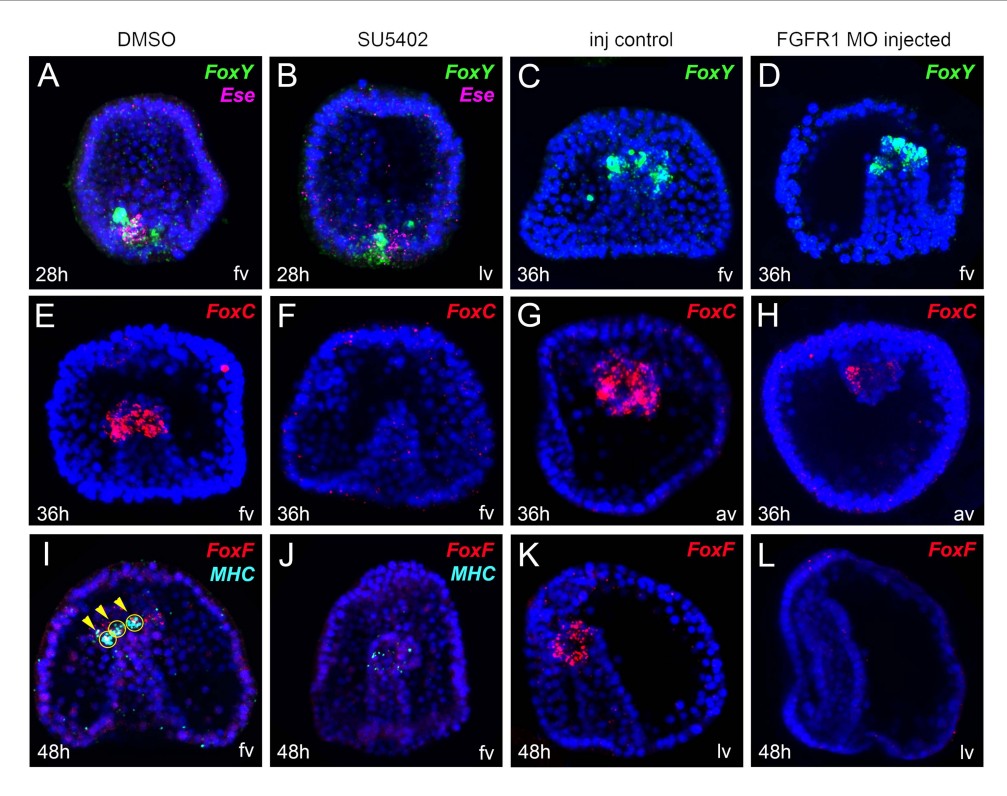

**Figure 3**. Spatial analysis of gene expression after FGF pathway perturbation by FISH. *FoxY* (**A–D**) *FoxC* (**E–H**), *FoxF* (**I–L**), *MHC* (**I**, **J**), and *Ese* (**A**, **B**) transcript localization tested by FISH in control embryos (**A**, **C**, **E**, **G**, **I**, **K**) and in embryos treated with SU5402 (**B**, **F**, **J**) or injected with FGFR1 MO (**D**, **H**, **L**) (for MO injection controls see also 'Materials and methods' and *Figure 2—figure supplement 2*). Panels **A**, **B**, **I**, and **J** show double FISH. *FoxY* was stained in green, *FoxC* and *FoxF* in red, *MHC* in cyan, and *Ese* in magenta. Nuclei were labeled blue with DAPI. Each picture is a stack of merged confocal Z sections. Yellow circles indicated by yellow arrowheads show cells co-expressing the analyzed genes. The orientation of the larvae is reported for each panel: fv, frontal view; av, animal view; lv, lateral view.

hand, *Ets1/2* expression showed a clear co-localization with *FoxC* transcripts at around 30 hr (*Figure 2—figure supplement 3E,F*) that perfectly corresponds to the onset of myogenesis, suggesting a putative role of Ets1/2 as one of the terminal effectors of MAPK pathway acting during myoblast specification. Further perturbation experiments are needed to confirm this hypothesis.

## Four members of the Forkhead family of transcription factors in the myogenic GRN model

Given the interesting temporal and spatial expression profile of *FoxY*, *FoxC*, *FoxL1*, and *FoxF* regulators during myogenesis and the decrease in *FoxC* and *FoxF* expression observed after perturbing FGF signaling, translation-blocking MOs specific for all four Fox factors were designed and tested at a concentration of 200 μM.

The injected embryos were fixed at the larval stage for immunostaining to test MHC protein levels. In 90% of the *FoxY* morphants, the archenteron was fully elongated with signs of incomplete differentiation and the coelomic pouches were lacking, as was shown previously (*Song and Wessel, 2012*; *Materna et al., 2013*). The MHC level was also severely reduced (*Figure 4A,B*). 80% of the *FoxC* morphants showed a similar morphological alteration; the archenteron was fully invaginated and partially compartmentalized, missing completely the coelomic pouches and MHC protein (*Figure 4C*). In 90% of the *FoxF* morphant larvae, a fully tripartite archenteron was present with disorganized coelomic pouches and a great decrease in the MHC level (*Figure 4D*). 95% of the *FoxL1* MO-injected larvae possessed a differentiated archenteron with disrupted coelomic pouches and reduced MHC protein level (*Figure 4E*). The phenotypes caused by FoxC, FoxF, and FoxL1 MOs were confirmed by injecting a second translation-blocking MO targeting to different sequences (*Figure 4—figure supplement 1*). These observations reinforce the crucial role of Fox family factors in sea urchin myogenesis.

## MyoD2, Tbx6, and Six1/2 are necessary for proper muscle development

MOs blocking the translation of three mesodermal markers *Tbx6*, *MyoD2*, and *Six1/2* that are known to have conserved roles in muscle development were included in the study. *MyoD2* and *Tbx6* are both parts of the molecular fingerprint of the myoblasts at the late gastrula stage, which makes them good candidates as myogenic regulators (*Andrikou et al., 2013*). *Six1/2* is known to be a part of the regulatory state of the aboral mesoderm, which gives rise to two other mesodermal derivatives (pigment cells and coelomic pouches) (*Poustka et al., 2007*). At the mid gastrula stage, *Six1/2* expression partially overlapped with that of *FoxC* in the oral mesodermal domain (*Figure 4—figure supplement 2*). This transient expression seems to be correlated with the transcription of a late isoform whose peak of expression was seen at 42 hr (*Figure 4—figure supplement 3A,B*). Therefore, a MO blocking the translation of the late isoform was designed and tested.

85% of the *MyoD2* morphants showed an elongated archenteron, and in most of the cases with reduced sphincter constrictions. The coelomic pouches were also not formed properly and the MHC level was dramatically reduced (*Figure 4F*). In 80% of the *Six1/2* morphants, the archenteron was elongated but not differentiated, the coelomic pouches were absent, and the MHC level was decreased (*Figure 4G*). Interestingly, unlike what previously shown by using an MO against the early isoform of *Six1/2*, which negatively affects pigmentation (*Ransick and Davidson, 2012*), embryos injected with the MO targeted to the late *Six1/2* isoform were fully pigmented (*Figure 4—figure supplement 3C,D*). Finally, 95% of *Tbx6* morphants had a milder phenotype, with no signs of malformed coelomic pouches and only a partial disrupted muscle fiber assembly as shown by MHC immunostaining (*Figure 4H*). Similar phenotypes were observed when injecting a second MO against *Six1/2* and *Tbx6* (*Figure 4—figure supplement 4*). The above experiments demonstrate an important role of MyoD2 and the late isoform of Six1/2 as well as a limited role of Tbx6 in sea urchin muscle development, thus, suggesting that Tbx6 acts in a redundant manner with other factors, a known property of T-box factors (*Gentsch et al., 2013*). To test this hypothesis, double knockdown assays should be conducted.

## Expression of selected mesodermal genes in *FoxY, FoxC, FoxL1, FoxF, Tbx6, MyoD2*, and *Six1/2* morphants

In order to identify the myogenic gene core set and unravel in detail the inter-regulatory mechanisms that take place during myogenesis, we tested the expression of genes encoding selected mesodermal factors and signaling components in the morphants by FISH (*Figure 5*),

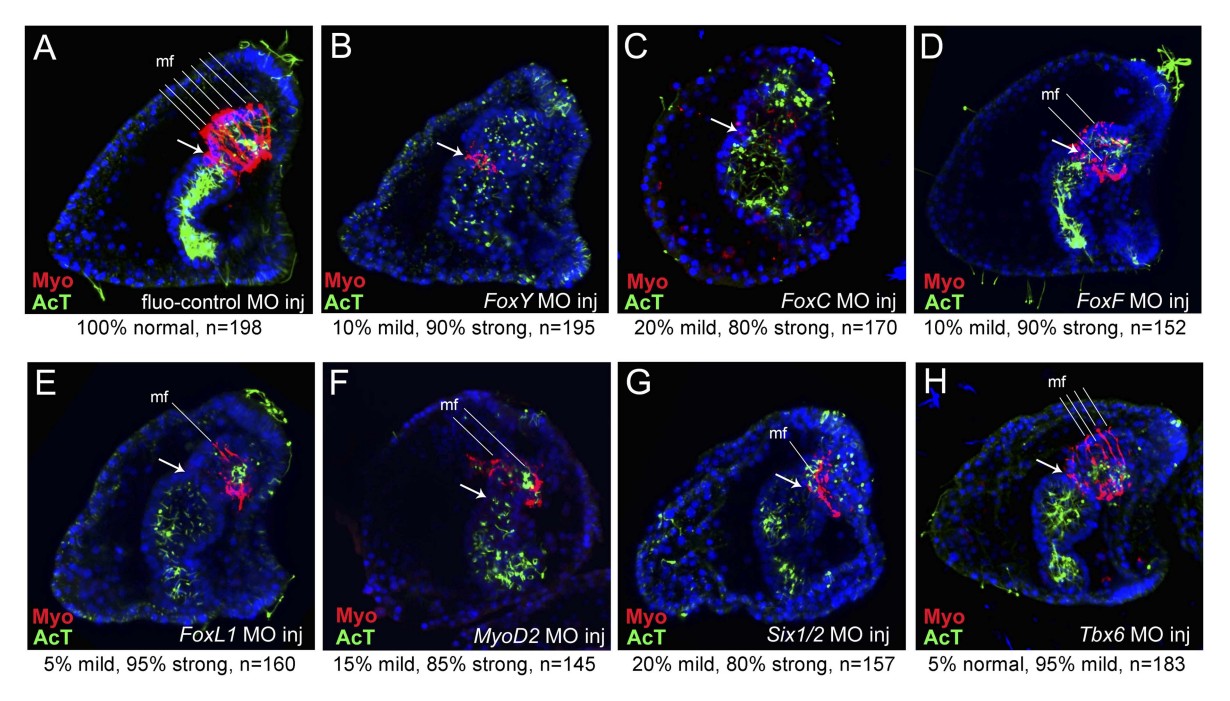

Figure 4. MHC protein detected by immunostaining after perturbation of putative myogenic regulators. MHC protein localization was tested by immunostaining in fluo-control MO-injected pluteus larvae (72 hr) (A) and in embryos of the same age injected with MOs against *FoxY* (B), *FoxC* (C), *FoxF* (D), *FoxL1* (E), *MyoD2* (F), *Six1/2N* (G), and *Tbx6* (H) (for MO injection controls see also Materials and methods and *Figure 2—figure supplement 2*). The ciliary band and gut internal cilia were stained by immunohistochemistry with an anti-acetylated tubulin antibody. Each picture is a stack of merged confocal Z sections with MHC in red and acetylated tubulin in green. Nuclei were labeled blue with DAPI. All embryos are seen in lateral view with the oral side on the right. White arrows indicate the position of cardiac sphincters. White lines indicate muscle fibers (mf). Below each panel, statistics of muscle fiber phenotype observed are reported as normal (6–7 mf), mild (4–5 mf), or strong (0–2 mf). A co-expression analysis of *Six1/2* and *FoxC* is reported in *Figure 4—figure supplement 1*. Analysis of the temporal expression profile of two distinct Six1/2 isoforms and visualization of pigmentation after perturbing Six1/2N isoform are reported in *Figure 4—figure supplement 2*.

The following figure supplements are available for figure 4:

Figure supplement 1. Control experiments for MOs.

Figure supplement 2. Co-expression analysis of *Six1/2* and *FoxC* by double FISH.

Figure supplement 3. The two *Six1/2* isoforms.

Figure supplement 4. Control experiments for MOs.

quantitative PCR (qPCR) (asterisks in *Figure 6*), and NanoString analysis (*Figure 6—source data 1, 2*). We selected three different developmental stages for our analyses that correspond to three distinct steps of myogenesis: early gastrula (30–35 hr) as myoblast specification stage, a broad mid gastrula (36–44 hr), and late gastrula (45–48 hr) stage as intermediate and later steps of myogenesis, respectively. For NanoString analyses, we used a code set containing probes for most known NSM factors expressed at these developmental stages and a number of signaling components. The MO effects on transcript levels analyzed using the three methods were mostly consistent. For quantitative analyses using QPCR and Nanostring (*Figure 6*), changes in the transcript level were considered significant if the effect between control and MO injected embryos is more than twofolds. Epistatic interactions were evaluated when the effects were observed in at least two significant data points.

At the early gastrula stage, *FoxY* and *FoxC* expression was strongly reduced in *FoxY* MO-injected embryos (*Figure 5A–D*), suggesting that *FoxY* positively regulates itself and *FoxC*. This positive

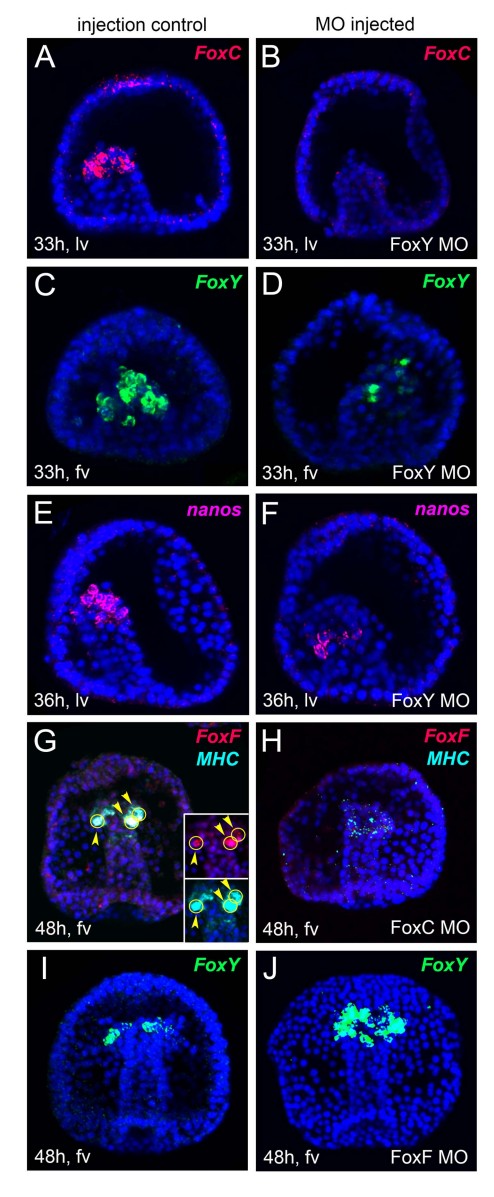

injection control    MO injected

**Figure 5**. Spatial analysis of gene expression after MO perturbation of selected putative myogenic regulators by FISH. *FoxC, FoxY, FoxF, MHC,* and *nanos* transcripts were detected by FISH in fluo-control MO injected embryos (**A**, **C**, **E**, **G**, **I**) and in embryos injected with MOs against *FoxY* (**B**, **D**, **F**), *FoxC* (**H**), and *FoxF* (**J**). All images are obtained as stacks of merged confocal Z sections. Panels **G**, **H** show double FISH. In panel **G**, single channels over DAPI are shown as insets. *FoxY* was stained in green, *FoxC* and *FoxF* in red, *MHC* in cyan, and *Nanos* in magenta. Nuclei were labeled blue with DAPI. Yellow circles indicated by yellow arrowheads show cells co-expressing the analyzed genes. The orientation of the embryos is indicated in each panel: fv, frontal view; lv, lateral view.

regulation continued to the mid and late gastrula stages and FoxY also positive regulated the other two Fox factors, *FoxF* and *FoxL1* (**Figure 6**). The expression of *Nanos,* a germ cell marker (**Juliano et al., 2006**) that is part of the molecular signature of the myoblasts precursors at that developmental time point (**Andrikou et al., 2013**), was also downregulated in *FoxY* morphants (**Figure 5E,F** and **Figure 6**). *FoxC, FoxF,* and *FoxL1* were downstream of *FoxC* at the mid gastrula stage. Expression of *FoxF* and *FoxL1* required *FoxC* input, whereas *FoxC* negatively regulated itself (**Figure 5G,H** and **Figure 6**). At the late gastrula stage, *FoxY* expression domain was expanded in the *FoxF* morphants to include the oral vegetal region of the archenteron tip, where myogenesis takes place (**Figure 5I,J**).

In addition to the inter-regulatory properties of the Fox factors, at the mid gastrula stage, *FoxY* also positively regulated *Pitx2* (**Hibino et al., 2006**), *MyoR,* a gene expressed in sea urchin mesoderm but not in the myoblasts (**Andrikou et al., 2013**), *Six1/2* and *ScratchX. FoxC* also activated *Tbx6* and repressed *Not* and *Dachshund (Dachs),* an aboral NSM gene in the sea urchin embryo (**Luo and Su, 2012**). *Six1/2* appeared to activate *FoxL1, MyoR,* and *Pitx2,* as previously suggested (**Hibino et al., 2006**). At the late gastrula stage, *FoxY* continued to activate the aforementioned downstream factors although its effect on *Six1/2* was diminished and activated in addition *SoxE* gene. *FoxL1* gave positive inputs to *FoxF, MyoR, ScratchX,* and *Pitx2* and repressed *Dachs, FoxY,* and *Not. Tbx6* repressed *FoxY, Dach,* and *Scl* and activated *Pitx2* and *MyoR.* Positive inputs on *Tbx6* gene from *FoxC* and on *MyoR* gene from *Six1/2* remained at the late gastrula stage. These perturbation analyses revealed complex positive and negative regulatory interactions among the four Fox factors and other transcriptional regulators, and the myogenic GRN models were formulated based on these results.

## A provisional GRN model driving muscle specification in the sea urchin embryo

This detailed perturbation analysis coupled with the available high-resolution transcriptional profiling during sea urchin embryogenesis (**Materna et al., 2010**) led to the construction of a GRN model that orchestrates sea urchin myoblast specification. Based on the examined developmental stages and regulatory states, three GRN diagrams have been illustrated using

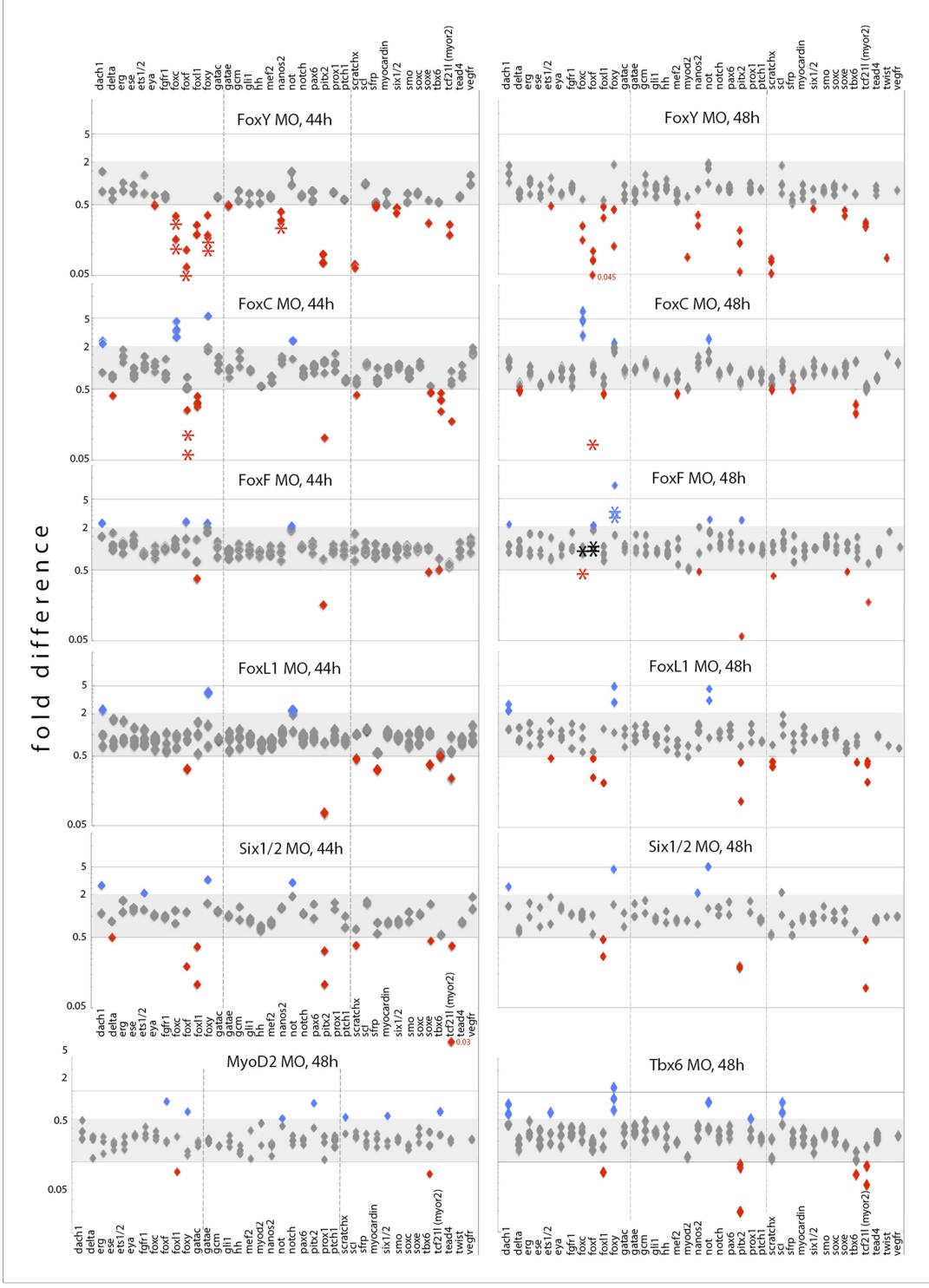

**Figure 6**. Effects of FoxY, FoxC, FoxF, FoxL1, MyoD2, Six1/2, and Tbx6 perturbations on transcript levels of selected mesodermal genes at 44 hr and 48 hr. Each diamond represents a single measurement of three independent biological experiments. Fold differences were calculated between experiments and control counts using the quantitative data obtained from the NanoString nCounter. Onefold change represents no change; ≥ 2 indicates increased expression level significantly (blue labels); ≤ 0.5 indicates decreased expression level significantly (red labels). Asterisks indicate perturbation effects as measured in independent biological experiments by qPCR. NanoString and qPCR perturbation data normalized against controls are provided in *Figure 6—source data 1*, and raw NanoString data are provided in *Figure 6—source data 2*.

*Figure 6. continued on next page*

*Figure 6. Continued*

The following source data are available for figure 6:

**Source data 1**. Perturbation data derived from NanoString and qPCR analysis showing fold differences of gene expression in MO-injected embryos after normalization against controls.

**Source data 2**. Raw data derived from NanoString analysis.

---

the BioTapestry software (www.biotapestry.org), one for the very early and two for the mid and late gastrula stages. The three proposed GRN models are summarized in *Figure 7*.

At the very early gastrula stage (28–32 hr; *Figure 7A*), FGFA is produced in the ventrolateral ectoderm and is received by FGFR1, which is expressed in all myoblast precursor cells. FGF signaling induces myoblast specification through the MAPK/ERK pathway and a downstream effector (indicated as a circle in *Figure 7A*), possibly Ets1/2, and results in the activation of *FoxC* transcription. *FoxC* activation needs an additional positive input from *FoxY*. Since *FoxY* expression is excluded from the other two NSM lineages (pigment and blastocoelar cells) (*Andrikou et al., 2013*), it is reasonable to consider the existence of a Non-Skeletogenic Mesoderm Repressor (NSMR) that inhibits *FoxY* transcription in these cell populations. The existence of such a repressor has also been postulated by *Materna and Davidson (2012)*. Here, we propose that in the myoblast precursors this repressive action is blocked through a double-negative gate, due to the existence of another repressor (X).

At the mid gastrula stage (40–44 hr; *Figure 7B*), *FoxY* establishes the myoblast regulatory state by activating a variety of transcriptional regulators including all four Fox factors. *FoxC* is a broad con-nected hub gene and gives positive inputs to a number of factors that compose the muscle gene battery such as *FoxF*, *FoxL1*, and *Tbx6*. *FoxL1* needs also an additional input from *Six1/2*. Finally, *SoxE* is also part of the molecular identity of the myoblasts at the mid gastrula stage since it appears to largely co-express with *FoxC* (*Figure 7—figure supplement 1A*); however, the upstream factor(s) controlling *SoxE* expression remain unknown.

At the late gastrula stage (44–48 hr; *Figure 7C*), the expression of *FoxY*, *Nanos*, *Six1/2*, and *SoxE* clears from the myogenic territory and confines in other NSM domains (small micromere [SM] and aboral NSM [AB NSM domain]). This occurs due to the repressive functions of *FoxF* and *Tbx6* on *FoxY*, which results in a subsequent loss of *Six1/2*, *SoxE*, and *Nanos* expression. Similarly, *Not* (whose expression domain relative to that of *FoxC* is reported in *Figure 7—figure supplement 1B*), as well as *Scl* and *Dach* receive negative inputs from *FoxL1* and *Tbx6*, respectively, that together prevent their expression in the myogenic domain. The initiation of *MHC* transcription marks the terminal differentiation state (*Andrikou et al., 2013*).

## Regulatory interactions in the other three NSM domains

The analyses of the perturbation data provided additional information concerning the regulatory interactions seen in the other three NSM domains. As elsewhere stated (*Materna et al., 2013*), *FoxY* is at the top of the hierarchy of the NSM GRN. *Tbx6* and *FoxL1* positively regulate *Pitx2* (whose expression domain relative to that of *FoxC* is reported in *Figure 7—figure supplement 1C*) and together with *Six1/2* activate *MyoR*. However, the fact that *Tbx6* and *FoxL1* are not expressed in the AB NSM (*Andrikou et al., 2013*) lead us to the conclusion that their inputs on *MyoR* are indirect. *FoxL1* may also activate *Pitx2* in the oral ectoderm (OR ECT). Finally, in the SM lineage, *FoxY* is upstream of *Nanos*, *FoxC*, and *Pitx2*.

## Discussion

### FGF signaling pathway triggers myoblast specification

In this study, we propose that the muscle progenitors originate from a pool of unspecified cells located at the oral/lateral periphery of the vegetal plate at the mesenchyme blastula stage and they adapt the myogenic fate by receiving an inductive signal at the moment of gastrulation. This would further imply that all developmental decisions regarding the separation of all four NSM regulatory

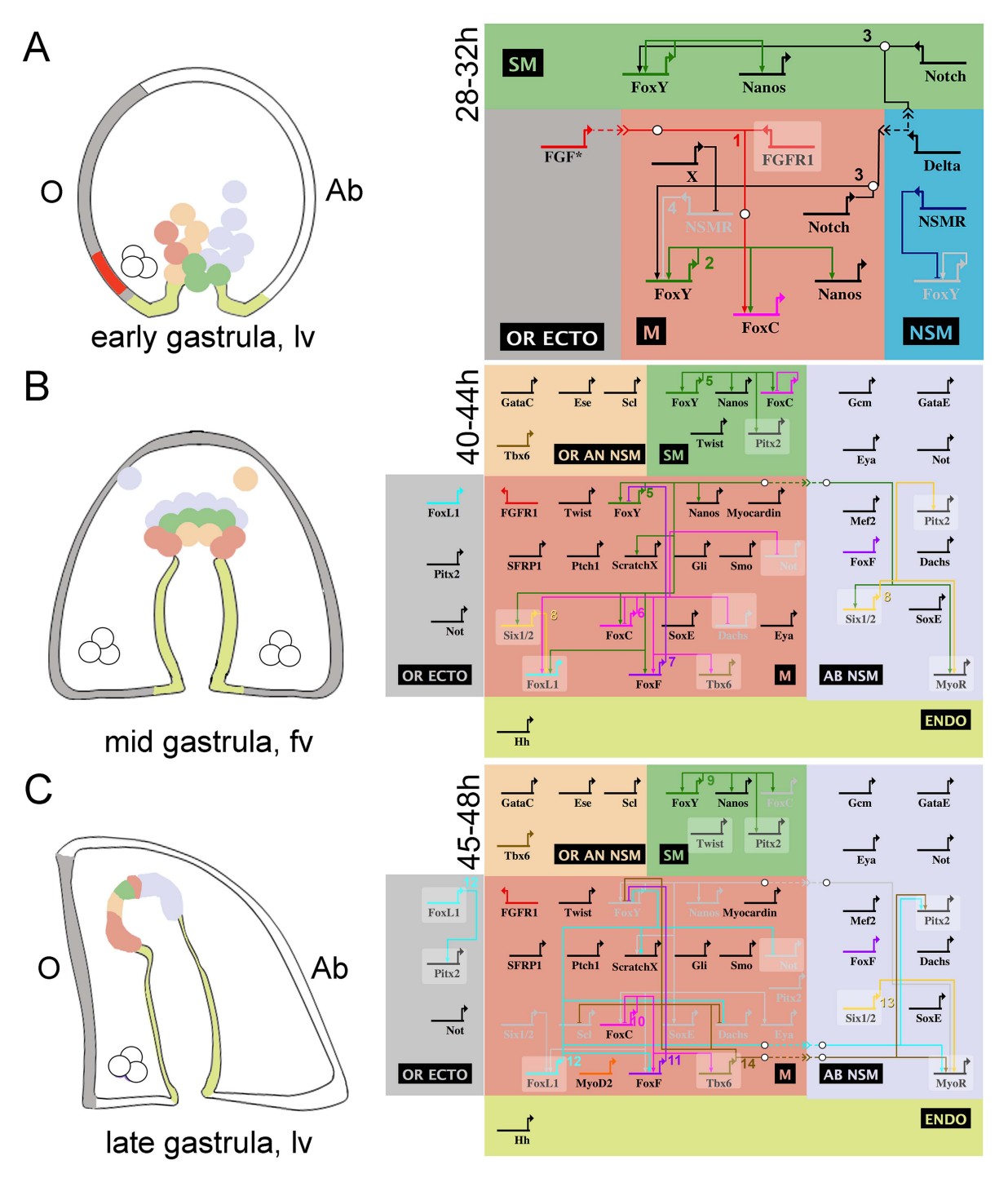

**Figure 7**. Schematic representation and view from all nuclei of the NSM regulatory interactions in early, mid, and late sea urchin gastrulae. On the left side, three developmental stages of the sea urchin embryo are schematized: (**A**) early, (**B**) mid, and (**C**) late gastrula stage. On the right side, the genetic interactions found within this study are summarized. Different colors are used for each domain showing exclusive regulatory states: oral animal non-skeletogenic mesodermal (NSM) (OR AN NSM), salmon pink; NSM, blue; aboral NSM (AB NSM), lavender; small micromere derivatives (SM), green; myogenic domain (M), light red; endoderm (ENDO), yellow-green; oral ectoderm (OR ECTO), light gray. Genes are presented as horizontal thick lines and their names are reported below the thick lines. The wiring among the genes is shown with solid lines, although none of them has been demonstrated to be direct. Arrows represent positive regulation, bars represent repression, and white bullets, together with the dashed lines, indicate signaling events. Genes that are expressed in more than one domain, for which the putative inputs were revealed by NanoString but not validated by spatial expression analysis, are shown on a shaded background. The asterisk in A relates to the fact

*Figure 7. continued on next page*

*Figure 7. Continued*

that we did not demonstrate which FGF factor signals to FGFR1. A co-expression analysis of several genes included in the gene regulatory network (GRN) diagrams is reported in *Figure 7—figure supplement 1*. Numbers associated to inputs indicate the evidence for all interactions reported and are listed in *Figure 7—source data 1*.

The following source data and figure supplement are available for figure 7:

**Source data 1**. Evidence for all inputs reported in *Figure 7*.

**Figure supplement 1**. Co-expression analysis of genes encoding mesodermal factors by double FISH.

states (pigment, blastocoelar, muscle, and coelomic pouch cells) take place during the interval between the blastula and the very early gastrula stage.

A key finding of this work was the recruitment of FGF signal in myoblast specification. FGF-signaling cascade is reported to trigger the emergence and/or promote the expansion of muscle lineage progenitors in different organisms and probably possessed an ancestral role in mesoderm patterning (at least in deuterostomes), along with other developmental processes. Genes encoding FGF and FGFR were already present in the last common ancestor of all metazoans, but the origin of the FGF/FGFR couple appears to be an innovation specific to the Eumetazoa, potentially linked to the increase of animal complexity (*Bertrand et al., 2014*). The acquisition of diverse roles and functions of FGF signaling occurred with the expansion of FGF and FGFR families during evolution (*Itoh and Ornitz, 2011*). For example, vertebrates are known to have the largest number of FGF-signaling components due to the two whole-genome duplication events (22 ligands and four receptors). FGF signals are involved in various developmental contexts (e.g., somitogenesis, cancer, gastrulation, metabolism, neural induction, etc) (*Dorey and Amaya, 2010*; *Oki et al., 2010*; *Naiche et al., 2011*; *Ko et al., 2014*; *Neugebauer and Yost, 2014*; *Sandhu et al., 2014*). Similar situations have been observed in *Drosophila* where the moderate expansions of both FGFR (*breathless* and *heartless*) and FGF (*Pyramus* [*Pyr*], *Thisbe* [*Ths*], and *Branchless* [*Bnl*]) families resulted in different biological functions such as cell–cell interactions during mesoderm layer formation, caudal visceral muscle formation, tracheal morphogenesis, and glia differentiation (*Muha and Muller, 2013*). In echinoderms, the two paralogues FGFR1 and FGFR2 are involved in two distinct developmental processes: myoblast specification (this study) and PMC migration (*Rottinger et al., 2008*), respectively. Diverse functions of FGF signaling seen among different animal taxa suggest that this signaling system is redeployed at different levels of GRN hierarchy and acquire new functions in developmental and physiological processes. Moreover, the hierarchical position of FGF signaling seems to depend on the level of complexity of the developmental process placing it as a 'checkpoint' in a non-conserved way regarding its downstream targets, a known property of GRN 'plug-in' devices defined previously (*Davidson, 2010*). The functional importance of being attachable onto any kind of circuitry reflects the need of developmental GRNs to adopt factors that act as turn on/off apparatuses, resulting in change of the undefined cell regulatory state to their specific cell lineage.

## The multilevel transcriptional regulation

An interesting finding of this analysis is the hierarchical organization of the regulatory interactions in the network topology. *FoxY*, a sea urchin-specific Fox family factor, lies on the top of the GRN architecture and is the key upstream regulator of sea urchin myogenesis and coelomic pouch formation. As shown elsewhere, *FoxY* is a direct target of the D/N signal, produced in the adjacent NSM. D/N signaling is known to be effective in cells being in direct contact with the ligand-producing cells (*Wang, 2011*). Therefore, the positive autoregulation of *FoxY* maintains its own expression during gastrulation when cells invaginate into the blastocoel and may be away from the signaling source.

*FoxY* then promotes the expression of downstream positive regulators necessary for the execution of the myogenic lineage fate. The next tier of the myogenic GRN includes *FoxC, SoxE, ScratchX*, and *Six1/2* factors and may be regarded as the core level of GRN. *FoxC* initially needs both positive inputs from FGF signaling and *FoxY* to be activated but then acts as an autorepressor in order to stabilize its number of transcripts. All factors belonging to this group are highly interconnected and provide multiple inputs to their downstream targets. On the same level of the GRN, immediately adjacent to

the core, stands the intermediate layer of the network composed of the *FoxF* and *FoxL1* transcriptional effectors. Finally, the last level of the network includes the differentiation driver and repressor *Tbx6* that triggers the terminal myoblast differentiation. *Tbx6* seems to cooperate redundantly with other factors in an OR logic (*Davidson, 2010*), which also explains the mild phenotype seen in the Tbx6 perturbations. The differentiation drivers, such as *MyoD* (*Andrikou et al., 2013*), are seen in the bottom of the GRN and probably activate the differentiation gene battery set, composed of a number of structural genes (e.g., *MHC*) that account for the specific function of the muscle cell.

This highly multilevel transcriptional regulation seen in myogenesis can be interpreted as an example of 'correlated evolution' where the increase of complex processes is accompanied by the expansion of the relevant regulatory systems whilst in less complex systems a shallower regulatory system can be provided.

## Recruitment of the Fox factors in the myogenic GRN

Another important outcome that arose from this analysis is the sequential inter-regulatory mechanism observed among the four Fox family members FoxY, FoxC, FoxF, and FoxL1. The Fox family of transcription factors is an ancient gene family and it has been proposed that its evolutionary origin occurred in a clade of unicellular organisms (*Baldauf, 1999*). This family has expanded over time through multiple duplication events, and sometimes through gene loss, and resulted in over 40 members in vertebrates, grouped in 23 subclasses (*Hannenhalli and Kaestner, 2009*). Among them, FoxC, FoxF, and FoxL subfamilies are of particular interest because they are clustered in most metazoan genomes and usually involved in mesoderm specification/differentiation processes (*Mazet et al., 2006*; *Shimeld et al., 2010*). Moreover, a linked activation or an overlapping expression has been reported in some cases, as in vertebrates, where FoxC specifies the dorsal mesoderm and derivatives, while FoxF patterns the lateral mesoderm and derivatives (*Beh et al., 2007*; *Zinzen et al., 2009*; *Amin et al., 2010*; *Fritzenwanker et al., 2014*). In the sea urchin, where a cluster of *FoxC*, *FoxF*, and *FoxL1* genes is present (*Andrikou et al., 2013*), we witness a similar regulatory logic where the different Fox factors pattern in an overlapping fashion in different compartments of the coelomic pouches. This cluster was probably expressed initially in developing mesodermal tissues and further evolved in regulating the specification and compartmentalization of mesodermal derivatives. Moreover, our findings highlight the importance of the hierarchical position of the Fox family factors in the GRN. The expression patterns and sequential activation of *FoxC*, *FoxF*, and *FoxL1* genes in time reflect the linkage properties of the retained cluster: among the three genes, *FoxC* is the first to be turned on in the myogenic lineage and is necessary for the activation of the downstream factors *FoxF* and *FoxL1*, which are inter-regulated. The outcome of such complex inter-wiring GRN may contribute to the establishment of a more robust output, able to mask putative perturbations of single nodes, as recently proposed (*Macneil and Walhout, 2011*).

## Exclusion of alternative NSM fates

This study provides additional insights into understanding the logic of the exclusive mechanisms that occur in the sea urchin embryo during myoblast specification (*Davidson, 2009*). By 24 hr post fertilization, the separation of the different NSM regulatory states is defined. The vegetal plate of the embryo consists of four distinct NSM cell populations: the aboral part that occupies about two-thirds of the total cells will differentiate into the pigment cells; the oral part that consists of the remaining one-third of the total cells will become the blastocoelar cells; the four SMs in the center are primordial germ cells; approximately 2–3 cells in the oral/lateral domain of the periphery of the vegetal plate will give rise to the myoblast precursors. As showed elsewhere, the specification of the distinct NSM domains depends on D/N and Nodal signals (*Sherwood and McClay, 1999*; *Ransick and Davidson, 2006*; *Duboc et al., 2010*). At 9 hr post fertilization, Delta ligand produced by the skeletogenic mesoderm activates a D/N cascade, which subsequently initiates *Gcm* and *GataE* transcription in all NSM cells (*Ransick and Davidson, 2006*; *Lee et al., 2007*). Later, at 24 hr, the NSM GRN becomes regionalized into distinct oral and aboral NSM GRNs in consequence of Nodal signaling through the immediate expression of a Nodal target, *Not*, in the oral NSM. This causes the repression of the aboral GRN in the oral NSM cells, and the aboral NSM cells still express *Gcm* and *Gata*E as part of the regulatory state of the pigment cell lineage (*Materna et al., 2013*). In the oral NSM, the expression of

a new suite of regulatory genes is taking place that belong to the blastocoelar lineage GRN (*Solek et al., 2013*). Our study showed that in the periphery of the oral/lateral NSM, the appearance of 2–3 unspecified cells is evident. These cells will soon get specified, at 28/30 hr, by receiving an inductive FGF signaling at the moment of gastrulation. FGF signal reception together with the double-negative gate caused by the repressive action of factor X on NSMR leads to the transcriptional activation of *FoxY* and *FoxC* expression. Positive inputs from *FoxY* and *FoxC* genes into downstream effectors promote the recruitment of the myogenic genes and lock down the myoblast regulatory state. Genes such as *Scl, Dachs, Pitx2,* and *Not* that belong to different NSM regulatory states are excluded from the myogenic domain leaving only the muscle GRN operating. Finally, spatial repression circuits generate regulatory transitions in the expression of key genes such as *FoxY, Nanos, SoxE,* and *Six1/2,* which now are established only in the other NSM GRNs. The sea urchin myogenic GRN is a nice example of how dynamic developmental processes can be encoded in the genome and shows clearly that understanding in depth the wiring properties of a developmental GRN model can provide a comprehensive view on the relationship between the regulatory architecture and gene expression dynamics.

### Rewiring the myogenic GRN over evolutionary time

The nature of the evolutionary alterations that arise from regulatory changes depends on the hierarchical positions of these changes within a GRN. One of the most striking findings of this study concerns the paradox that genes are constantly re-used in the same context, but are rewired in different networks. Despite the conserved regulatory modules found in the system, the myogenic GRN structure has diverged extensively among animal groups. As a consequence, the level of the functional importance of the homologous transcriptional regulators (e.g., *Six1/2* in sea urchin and *Ceh-34* in *C. elegans* or MyoD in vertebrates and *Nautilus* in *Drosophila*) (this study, [*Olson and Klein, 1994*; *Balagopalan et al., 2001*; *Amin et al., 2009*]), as reflected from the position within the GRN, is often diversified. New, lineage-specific genes are recruited (*FoxY* in sea urchin), and 'master regulatory genes' either have been lost completely (e.g., the absence of *Pax3/7* ortholog in the sea urchin) or lose their hierarchical position within the network (e.g., *Nautilus* in *Drosophila* and *MyoD* in vertebrates), with the level of complexity to be reflected in the wiring density and in the GRN organization. However, the fact that the same factors are used over and over again in such different animal systems indicates that the modular components are somehow required for keeping their myogenic activity during evolutionary time. It seems that as transcription factor families expanded and functionally diversified during evolution, the ancestral myogenic function may have been preserved in a more distant family member, rather than the homologous gene, providing the system with several regulatory alternatives, and explaining the high degree of evolutionary plasticity of developmental GRN architecture (*Andrikou and Arnone, 2015*).

   In conclusion, this large scale GRN analysis demonstrated a necessary hierarchical role for a large number of transcriptional regulators in muscle development and explained in a rational way the core gene network that is orchestrating the specification of the myogenic lineage. Moreover, it revealed the key signaling events involved in the activation of the muscle gene battery and underlined their crucial role in transforming an unspecified cell into a specific cell type with a characteristic molecular signature. Finally, this study reinforces the importance of GRN-based approach in understanding in detail complex developmental processes by assessing the causality of the regulatory mechanisms that accompany each step of the process.

## Materials and methods

### Animal husbandry and embryo cultures

Adult *S. purpuratus* were obtained from Patrick Leahy (Kerckhoff Marine Laboratory, California Institute of Technology, Pasadena, CA, USA) and housed in circulating seawater aquaria in the Stazione Zoologica Anton Dohrn of Naples. Spawning was induced by vigorous shaking of animals or by intracoelomic injection of 0.5 M KCl. Embryos were cultured at 15℃ in Millipore filtered Mediterranean seawater diluted 9:10 (Vol:Vol) in deionized $H_2O$.

## Perturbation experiments

FoxC, FoxF, FoxL1, MyoD2, Tbx6, FGFR1, and Six1/2 antisense MOs were obtained from Gene Tools (Pilomath, OR, USA) and injected at different concentrations in the presence of 0.12 M KCl. Various MO concentrations were tested and the lowest that enabled the observation of a phenotype was used for the experiments (500 µM for the FGFR1 MO and 200 µM for FoxC, FoxF, FoxL1, MyoD2, Tbx6, and Six1/2). Second MOs for FoxC, FoxF, FoxL1, Six1/2, and Tbx6 were used to confirm the morphant phenotypes (*Figure 4—figure supplements 1, 4*). As a control experiment, a Standard Morpholino Control oligo end modified with 3′-Carboxyfluorescein (control-fluo MO, Gene Tools) was injected in parallel at the same concentration as the corresponding experiments (*Figure 2—figure supplement 2*). Embryos injected with FoxY, FoxC, FoxF, FoxL1, MyoD2, Tbx6, and Six1/2 MOs displayed a normal gross morphology, similar to uninjected or fluo-control MO injected embryos up to the pluteus stage, except for the effects on the coelomic pouches, and in the case of FoxC MO, the apical organ, thus, suggesting confined effects of these MOs in the expression domains of the corresponding targeted genes. To test FGFR1 MO efficacy, embryos were injected with mRNA containing the MO target sequence fused to the 5′ of the *gfp*-coding sequence (500 ng/µl) with or without the MO (*Figure 2—figure supplement 2*). FoxY MO was kindly provided by Stefan Materna (Caltech, USA). MO sequences used in this study are listed in *Supplementary file 1*.

## Chemical treatments

SU5402 was dissolved in DMSO and added to a final concentration of 20 µM at 26 hr, 28 hr, 30 hr, or 36 hr up to the collection time. Higher concentrations than this were lethal to the embryos soon after the addition of the drug and addition of the drug after 30 hr did not show any effect. U0126 was dissolved in DMSO and added to a final concentration of 10 µM at 24 hr as reported previously (*Fernandez-Serra et al., 2004*). A corresponding volume of DMSO was added as controls. A table summarizing the drug treatments and the observed phenotypes is seen in *Figure 2—figure supplement 1*.

## PCR cloning and construction of expression plasmids

The primers used to amplify FGFA from embryonic cDNA were designed based on the gene models. 5′ and 3′ sequences were extended by the FirstChoice RLM-RACE Kit (Ambion, Austin, TX, United States). The complete mRNA sequences were deposited into GenBank (Sp-FGFA, HQ107979). FGFR1 was amplified based on the published sequence (U17164). Both primer sets are in *Supplementary file 1*. To construct the dominant-negative form of FGFR1, a DNA fragment containing the signal peptide, the extracellular, and transmembrane domains was amplified by PCR (forward primer: 5′-CGGGATCCATGAGTCTGCCGCGTTGTCC-3′, reverse primer: 5′-CCATCGATTGT CTCGAGGGAACTCCCAC-3′) and cloned into the pCS2+MT vector.

## Whole-mount in situ hybridization

In situ RNA probe sequences for *FoxY, FoxC, FoxF, Ese, Nanos, Six1/2, SoxE, MHC*, and *Gcm* are as previously published (*FoxY*: [*Ransick et al., 2002*]; *FoxC, FoxF*: [*Tu et al., 2006*]; *Ese*: [*Rizzo et al., 2006*]; *Nanos*: [*Juliano et al., 2006*]; *Gcm*: [*Ransick et al., 2002*]; *Six1/2, SoxE*, and *MHC*: [*Andrikou et al., 2013*]). Labeled probes were transcribed from linearized DNA using digoxigenin-11-UTP or fluorescein-12-UTP (Roche, Indianapolis, IN, USA), or labeled with DNP (Mirus, Madison, WI, USA) following kit instructions. For whole-Mount In situ hybridization (WMISH) with single probe, we followed the protocol outlined in (*Minokawa et al., 2004*). Double FISH and immunohistochemistry coupled to FISH were performed as described in (*Andrikou et al., 2013*). For triple FISH, the third signal was developed using a 488 fluorophore-conjugated tyramide (Invitrogen). Embryos were imaged with a Zeiss Axio Imager M1. FISHs were imaged with a Zeiss 510Meta confocal microscope.

## Whole-mount immunohistochemistry

Embryos were collected by gentle centrifugation and fixed in 2% paraformaldehyde in PBS for 15 min, washed 3 times in PBST, and incubated in 4% sheep serum and 1 mg/ml BSA in PBST for 30 min. Embryos were then incubated with a primary antibody (anti-*Sp*MHC, rabbit polyclonal antibody, diluted 1:600, PRIMM, Italy or a commercially available anti-P-Elk1 [Serine 383], mouse monoclonal antibody, dilution 1:100, Santa Cruz Biotechnology, Santa Cruz, USA) overnight at 4°C,

washed 4 times in PBST, and followed by another incubation in 4% sheep serum and 1 mg/ml BSA in PBST for 30 min. Similarly, embryos were then incubated with a secondary antibody (anti rabbit-AlexaFluor 555, Invitrogen or anti mouse-HRP) diluted 1:1000 for 1 hr in RT, washed 4 times in PBST, and imaged with a Zeiss 510Meta confocal microscope.

## RNA extraction and transcriptional profiling

Total RNA was isolated from cultures of various embryonic stages, approximately 100 embryos per replica. The RNA was extracted with RNAquous (Ambion). The samples were treated with DNase I (Ambion) to remove DNA contamination as described by the manufacturer. First-strand cDNA was synthesized from total RNA using the VILO kit (Invitrogen) according to the manufacturer's protocol. Expression levels were quantified using the NanoString nCounter (NanoString, UCL, London) with a custom-designed probe set of 40 genes (*Supplementary file 1*). Samples were processed according to manufacturers' instructions and data processed as described previously (*Materna and Davidson, 2012*). Thresholds of 2- and 0.5-fold differences were chosen as significant changes (*Materna and Davidson, 2012*). Some data points were supplemented with results from quantitative real-time PCR (qPCR) analyses. qPCR was conducted as described (*Rast et al., 2000*), using the ViiA 7 REAL TIME PCR detection system and SYBR green chemistry (Applied Biosystems, Foster City, CA, USA). The primer sequences used are included in *Supplementary file 1*. ddCt values were calculated between experiment and control embryos and converted to fold differences to be comparable to the NanoString data. Fold changes were calculated using *poly-ubiquitin* as a reference, and a threshold of twofold difference was chosen as a significant change (*Materna and Oliveri, 2008*). Normalized perturbation data against control are reported in *Figure 6—source data 1*, and raw data are reported in *Figure 6—source data 2*.

## Sequence and phylogenetic analysis

The signal peptide cleavage site of FGFA was predicted by SignalP 3.0 (http://www.cbs.dtu.dk/services/SignalP/). The core sequences of FGFs from different organisms were aligned using the Clustal W program, and the alignment was confirmed manually. After removing gaps, the verified alignments were used to construct phylogenetic trees with the MacVector software based on the neighbor-joining method. Bootstrap support values were calculated by 1000 pseudoreplications. All phylogenetic trees were illustrated with the FigTree program (http://tree.bio.ed.ac.uk/software/figtree/). The phylogenetic tree for FGFRs was constructed in the same manner as the FGF tree but was based on the alignments of the tyrosine kinase domains.

## Acknowledgements

The authors would like to thank Drs. Andy Ransick, Stefan Materna, and Paola Oliveri for kindly donating some of the reagents and clones used in this study, Paola Oliveri also for helping with the NanoString analysis and for useful discussions. Aaron Tolwson and Claudia Cuomo for their help with qPCR. Giovanna Benvenuto for support with confocal microscopy. Han-Ru Li for her help in morpholino injections and phalloidin staining.

## Additional information

### Funding

| Funder | Grant reference | Author |
| --- | --- | --- |
| European Commission (EC) | Marie Curie ITN project 215781 | Carmen Andrikou, Maria Ina Arnone |
| POR Campania FSE 2007-2013, Italy | project MODO | Carmen Andrikou, Maria Ina Arnone |
| Ministry of Science and Technology, Taiwan (MOST) | 101-2923-B-001-004-MY2 | Yi-Hsien Su, Maria Ina Arnone |
| Ministry of Science and Technology, Taiwan (MOST) | 103-2311-B-001-020-MY3 | Yi-Hsien Su |

| Funder | Grant reference | Author |
| --- | --- | --- |
| Ministry of Science and Technology, Taiwan (MOST) | 103-2627-B-001-001 | Yi-Hsien Su |

The funders had no role in study design, data collection and interpretation, or the decision to submit the work for publication.

### Author contributions

CA, Conception and design, Acquisition of data, Analysis and interpretation of data, Drafting or revising the article; C-YP, Acquisition of data; Y-HS, Analysis and interpretation of data, Drafting or revising the article, Contributed unpublished essential data or reagents; MIA, MIA, Conception and design, Analysis and interpretation of data, Drafting or revising the article

## Additional files

### Supplementary file

• Supplementary file 1. Primers, MOs, and nanoprobe sequences.

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
