## [Decision Letter]

Thank you for sending your work entitled “Logics and properties of a genetic regulatory program that drives embryonic muscle development in an echinoderm” for consideration at *eLife*. Your article has been favorably evaluated by Diethard Tautz (Senior Editor), a Reviewing Editor, and two peer reviewers.

The Reviewing Editor and the reviewers discussed their comments before reaching their decision, and the Reviewing Editor has assembled the following comments to help you prepare a revised submission.

To be acceptable for publication the manuscript requires substantial revision. We recognise that it represents a large amount of work and appreciate the new insights that it provides into the GRN underlying entry into myogeneis in the sea urchin model. However, the results are over-interpreted and parts of the data are not sufficiently validated for inclusion in the GRN, which requires at least one repeat experiment and preferably by an independent test. This is particularly the case for Hh and SFRP/Wnt signaling as well as Pitx2 and other factors. The FGF data are also unsatisfactory since it is not clear which Ets factor mediates the signal. It is also not clear why, based on the authors criteria for inclusion, some genes are excluded. Why for example is FoxC included and FoxF excluded. A detailed list of data points to be re-examined is given at the end of this letter.

Another concern of the reviewers is the morpholino data and the controls. It would be more convincing if more than one morpholino was shown to give the same phenotype. Knockdown of a GFP expressing construct, containing a sequence recognised by the morpholino is not sufficient and the use of a control morpholino simply shows that toxicity is not a problem for a batch of embryos.

Overall the key nodal points should be presented based on solid validation and everything else should be eliminated, with a discussion that now focusses on the key points instead of an extensive and confusing discussion of weak data. We hope that you can modify the manuscript along these lines.

The following concerns have to be dealt with:

1) If you apply the rule of more data points being necessary for inclusion, what else would fall out? Or, what might be included that is not currently included?

FoxY -> SoxE which also is projected in one location where SoxE is located but not in the other, yet the RNA you obtain includes both regions.

FoxY ->Gatae; FoxC ->SoxE; FoxY->SoxE; Six1/2-|Not; FoxL1-|FoxY

FoxL1->Pitx2 which you don't include but it is significant by your rules. So, why is this one not included when the others above it are? By the rule of more than one it would be as it is, not included.

FoxL1->Srfp

FoxL1->MyoR is included though your data doesn't show significance by your rules, and only one data point is significant by the rule of more than 2 fold.

FoxF-|Not is included even though it doesn't reach significance by your rules and also doesn't reach significance by the rule of needing more than one data point.

FoxF->Pitx2 is not included even though by your rules it should be, but by the rule of more than one, it should not.

FoxC->FoxF is included even though it should not be by your rules, and the one data point you have is supplied by the in situ. If the in situ is to be accepted there should be an indication that what is shown is not simply “best case” but a consistent finding.

FoxY->FoxL1 is not included though by your rules it should be and it also should be by the rule of more than one.

FoxY->MyoR is not included yet is one of the strongest knockdown effects you see.

All told, you model 29 connections in the 44 hr GRN, yet there are 14 questionable connections due to only one data point without confirmation, or by not including a connection according to your rules when a point should be included by those rules.

2) Situations exist where a perturbation has a very strong effect in the above, or at least a consistent effect, yet you don't include that connection. For example FoxY knockdowns have a very strong effect on expression of FoxF at 44 hrs – even stronger, relatively, than on FoxC – yet FoxC is included and FoxF is not. You make the call that FoxC has an input into FoxF and could argue that it is through that connection that FoxF is knocked down in FoxY MO. Though you have a connection there, your FoxC knockdowns don't show significance in knocking down FoxF. Since we are looking for consistency in the calls, this doesn't make sense.

3) For the 48 hr GRN the following data points do not satisfy your rules, or fail if a requirement of more than one data point is included.

MyoD-|Pitx2

FoxF-|FoxY

FoxF->MyoD

FoxF-| Not is included though is not significant by your rules.

FoxY->FoxL1 is not included though by your rules it should be.

FoxY->MyoD is not included though by your rules it should be; by the rule of needing more than one data point it should not be.

FoxY->Scratch is not included though by your rules it should be.

FoxY->MyoR is not included though by your rules it should be.

FoxL1-|FoxY not included though should be by your rules.

FoxL1->Pitx2 not included though should be by your rules.

FoxL1-|Not is included though goes against your rules.

Six1/2-|Dach is not included though by your rules it should be; by the rule of more than one data point it should not be.

Six1/2-|FoxY is not included though by your rules it should be. By rule of more than one it should not be.

Six1/2-|Not not included though by your rules it should be. By rule of more than one it should not be.

Tbx6->Pitx2 is not included though by your rules it should be.

Tbx6->MyoR is not included though by your rules it should be.

FoxY->twist is included though only one data point.

In this 45-48 hr GRN model, of 26 connections modeled, 17 connections are in question. 11 of those 17 are connections not included though by your rules they should be. It appears that even though you include all the data, your model is selective by not adhering to a set of rules. You have to apply consistent rules, and we strongly suggest for confidence that you adhere to a rule where more than one data point must be necessary to have some confidence that a call is likely to be correct.

---

## [Author Response]

*To be acceptable for publication the manuscript requires substantial revision. We recognise that it represents a large amount of work and appreciate the new insights that it provides into the GRN underlying entry into myogeneis in the sea urchin model. However, the results are over-interpreted and parts of the data are not sufficiently validated for inclusion in the GRN, which requires at least one repeat experiment and preferably by an independent test. This is particularly the case for Hh and SFRP/Wnt signaling as well as Pitx2 and other factors. The FGF data are also unsatisfactory since it is not clear which Ets factor mediates the signal. It is also not clear why, based on the authors criteria for inclusion, some genes are excluded. Why for example is FoxC included and FoxF excluded. A detailed list of data points to be re-examined is given at the end of this letter*.

We thank the editors and reviewers for their constructive suggestions. We have reinterpreted the data in a consistent way as suggested. We also rewrote the part about FGF/Ets and its downstream factors. Detailed explanation specific to each point is addressed below.

*Another concern of the reviewers is the morpholino data and the controls. It would be more convincing if more than one morpholino was shown to give the same phenotype. Knockdown of a GFP expressing construct, containing a sequence recognised by the morpholino is not sufficient and the use of a control morpholino simply shows that toxicity is not a problem for a batch of embryos*.

We have used second anti translation morpholinos targeted to FoxC, FoxF, FoxL1, Six1/2, and Tbx6 to validate our results (see Figure 4—figure supplement 1 and Figure 4—figure supplement 4 and [Supplementary-material SD4-data]). The same phenotype (reduction of the muscle fibers) has been observed when the second morpholinos were injected into the embryos.

*Overall the key nodal points should be presented based on solid validation and everything else should be eliminated, with a discussion that now focusses on the key points instead of an extensive and confusing discussion of weak data. We hope that you can modify the manuscript along these lines*.

*The following concerns have to be dealt with*:

1) If you apply the rule of more data points being necessary for inclusion, what else would fall out? Or, what might be included that is not currently included?

We have applied a new rule in which at least two data points showed more than two-fold changes (<0.5 or >2 fold difference) would be included in the network. The data are obtained in three different ways, including Nanostring measurements, QPCR analyses, and in situ hybridization.

*FoxY -> SoxE which also is projected in one location where SoxE is located but not in the other yet the RNA you obtain includes both regions*.

According to the new rule, we have removed the link from FoxY to SoxE at 44 h but added such link in both domains at 48 h network.

FoxY ->Gatae

The link has been removed.

FoxC ->SoxE

The link has been removed.

FoxY->SoxE

The link has been removed.

Six1/2-|Not

The link has been removed.

FoxL1-|FoxY

We have removed the link at 44 h but added such link at 48 h network.

FoxL1->Pitx2 which you don't include but it is significant by your rules. So, why is this one not included when the others above it are? By the rule of more than one it would be as it is, not included.

We have removed the link based on the new rule.

FoxL1->Srfp

The link has been removed.

*FoxL1->MyoR is included though your data doesn't show significance by your rules, and only one data point is significant by the rule of more than 2 fold*.

The link has been removed.

*FoxF-|Not is included even though it doesn't reach significance by your rules and also doesn't reach significance by the rule of needing more than one data point*.

We have removed the link based on the new rule.

*FoxF->Pitx2 is not included even though by your rules it should be, but by the rule of more than one, it should not*.

We have removed the link based on the new rule.

*FoxC->FoxF is included even though it should not be by your rules, and the one data point you have is supplied by the in situ. If the in situ is to be accepted there should be an indication that what is shown is not simply “best case” but a consistent finding*.

We have performed new experiments using QPCR to confirm the link (see [Supplementary-material SD1-data] and Figure 6). Therefore, based on in situ data and QPCR, we have validated the link.

*FoxY->FoxL1 is not included though by your rules it should be and it also should be by the rule of more than one*.

We have added the link in both 44 and 48 network.

*FoxY->MyoR is not included yet is one of the strongest knockdown effects you see*.

We have added the link in both 44 and 48 network.

*2) Situations exist where a perturbation has a very strong effect in the above, or at least a consistent effect, yet you don't include that connection. For example FoxY knockdowns have a very strong effect on expression of FoxF at 44 hrs – even stronger, relatively, than on FoxC – yet FoxC is included and FoxF is not. You make the call that FoxC has an input into FoxF and could argue that it is through that connection that FoxF is knocked down in FoxY MO. Though you have a connection there, your FoxC knockdowns don't show significance in knocking down FoxF. Since we are looking for consistency in the calls, this doesn't make sense*.

We have added the link from FoxY to FoxF at 44 h. The link from FoxC to FoxF at 44 h is retained because in addition to one supporting data from Nanostring measurement, two biological replicates analyzed by QPCR showed strong effects.

*3) For the 48 hr GRN the following data points do not satisfy your rules, or fail if a requirement of more than one data point is included*.

MyoD-|Pitx2

The link has been removed.

FoxF-|FoxY

This link is retained because, in addition to one supporting data from Nanostring measurement, two biological replicates analyzed by QPCR showed strong effects.

FoxF->MyoD

The link has been removed.

*FoxF-| Not is included though is not significant by your rules*.

The link has been removed.

*FoxY->FoxL1 is not included though by your rules it should be*.

We have added the link.

*FoxY->MyoD is not included though by your rules it should be; by the rule of needing more than one data point it should not be*.

The link has been removed because only one data point supported.

*FoxY->Scratch is not included though by your rules it should be*.

We have added the link.

*FoxY->MyoR is not included though by your rules it should be*.

We have added the link.

*FoxL1-|FoxY not included though should be by your rules*.

We have added the link.

*FoxL1->Pitx2 not included though should be by your rules*.

We have added the link.

*FoxL1-|Not is included though goes against your rules*.

The link is retained because two of the three biological replicates measured by Nanostring showed strong effects.

*Six1/2-|Dach is not included though by your rules it should be; by the rule of more than one data point it should not be*.

By the rule of more than one data point we decided not to include the link.

*Six1/2-|FoxY is not included though by your rules it should be. By rule of more than one it should not be*.

By the rule of more than one data point we decided not to include the link.

*Six1/2-|Not not included though by your rules it should be. By rule of more than one it should not be*.

By the rule of more than one data point we decided not to include the link.

*Tbx6->Pitx2 is not included though by your rules it should be*.

We have added the link.

*Tbx6->MyoR is not included though by your rules it should be*.

We have added the link.

*FoxY->twist is included though only one data point*.

We have removed the link.

*In this 45-48 hr GRN model, of 26 connections modeled, 17 connections are in question. 11 of those 17 are connections not included though by your rules they should be. It appears that even though you include all the data, your model is selective by not adhering to a set of rules. You have to apply consistent rules, and we strongly suggest for confidence that you adhere to a rule where more than one data point must be necessary to have some confidence that a call is likely to be correct*.

We thank the editors and reviewers for the great suggestion. We have now adhered to the rule of at least two data points must be necessary to have the links being included in the network.